# State-of-the-Art Internet of Things in Protected Agriculture

**DOI:** 10.3390/s19081833

**Published:** 2019-04-17

**Authors:** Xiaojie Shi, Xingshuang An, Qingxue Zhao, Huimin Liu, Lianming Xia, Xia Sun, Yemin Guo

**Affiliations:** 1School of Agricultural Engineering and Food Science, Shandong University of Technology, No. 12, Zhangzhou Road, Zibo 255049, Shandong, China; sxj6198@163.com (X.S.); axsoffice@126.com (X.A.); zqx5230@163.com (Q.Z.); liuhuimin1210@126.com (H.L.); 2Shandong Provincial Engineering Research Center of Vegetable Safety and Quality Traceability, No.12 Zhangzhou Road, Zibo 255049, Shandong, China; 3Zibo City Key Laboratory of Agricultural Product Safety Traceability, No.12 Zhangzhou Road, Zibo 255049, Shandong, China

**Keywords:** Internet of things, protected agriculture, integrated application, state-of-the-art

## Abstract

The Internet of Things (IoT) has tremendous success in health care, smart city, industrial production and so on. Protected agriculture is one of the fields which has broad application prospects of IoT. Protected agriculture is a mode of highly efficient development of modern agriculture that uses artificial techniques to change climatic factors such as temperature, to create environmental conditions suitable for the growth of animals and plants. This review aims to gain insight into the state-of-the-art of IoT applications in protected agriculture and to identify the system structure and key technologies. Therefore, we completed a systematic literature review of IoT research and deployments in protected agriculture over the past 10 years and evaluated the contributions made by different academicians and organizations. Selected references were clustered into three application domains corresponding to plant management, animal farming and food/agricultural product supply traceability. Furthermore, we discussed the challenges along with future research prospects, to help new researchers of this domain understand the current research progress of IoT in protected agriculture and to propose more novel and innovative ideas in the future.

## 1. Introduction

The Internet of Things (IoT) was first developed in 1999 by the network radio frequency identification (RFID) system proposed by the Massachusetts Institute of Technology (MIT) Auto-ID Labs [1]. With the application and development of new information technologies, the connotation and extension of IoT have undergone major changes [2]. IoT can be defined as a huge internet-based network connecting physical and virtual “things” with standard and interoperable communication protocols. To be specific, everything, such as a sensor and an actuator possessing unique identity and attribute, exchange messages and communicate with each other to realize intelligent positioning, tracking, identification, perception, monitoring and management via kinds of networks anytime and anywhere [3]. IoT has penetrated pervasively most aspects of human life everywhere such as health care, smart home, smart city, industrial control and so on. Agriculture is an ideal candidate for the deployment of IoT solutions because it occurs in wide areas that need to be continuously monitored and controlled [4,5].

The concept of protected agriculture is relative to open-field agriculture. It uses artificial techniques to change climatic factors such as natural light, temperature and humidity to create environmental conditions suitable for the growth of animals and plants, enabling them to grow around the clock. The environment of protected agriculture is completely or largely artificially controlled andwhich has broken the limits of climate and land conditions for the growth of animals and plants to a certain extent. So it is also called controllable agriculture. Compared with open-field agriculture, protected agriculture has more potential to apply IoT technology, because it is less affected by climatic and geographical factors. Some mature IoT solutions in other fields can be transferred directly to protected agriculture [6].

With the development of agricultural sensor, wireless communication, cloud computing, machine learning and Big Data technologies, IoT technology has emerged and is gradually being promoted and applied in the protected agriculture field [7,8,9]. It is playing an important role in various areas of protected agriculture as it is capable of helping farmers monitor soil condition, climate change and animal and plant health [10]. When the environmental factor change beyond the set threshold, IoT will automatically send a warning message to the administrator to remove the hidden danger. It can also control environmental factors such as temperature, humidity, carbon dioxide concentration and illumination according to the condition of crop growth in real time [11,12]. In addition, cameras in the IoT system are able to capture crop diseases and insect pests in the greenhouse in real time, helping farmers find problems and take targeted preventive measures [13]. Through GPS, RFID and other location-based sensors, goods such as vegetables can be tracked and monitored visually during transportation and storage. Supermarket managers use smart phone or PC to monitor and predict product status and the demand for getting product on shelves. At the user or consumer end, people can query the variety, origin, processing and other farm product information by means of the QR code, barcode, etc. IoT for protected agriculture can help create an informed, connected, developed and adaptable rural community. Low-cost embedded devices can improve the interaction between humans and the physical world. Cloud computing, edge computing, and Big Data can provide valuable analysis and support for decisions. In summary, IoT will become an important tool in the next few years to engage people in embedded agriculture which includes suppliers, farmers, technicians, distributors, merchants, consumers and government representatives [14,15].

On the basis of the potential of IoT applications in protected agriculture described in the previous paragraphs, this paper aims to identify the current state of solutions in these fields. The IoT fundamental structure in protected agriculture is introduced in detail based on agricultural sensors, wireless sensor networks, cloud computing, edge clouding and machine learning technologies. The application of IoT in protected agriculture and how it makes contributions is displayed. Furthermore, the open issues, challenges, future opportunities and development trends are discussed.

In this article, a comprehensive and profound review of IoT in protected agriculture has been executed based on a survey of real-world deployments and published papers over the past 10 years. Firstly, a keyword-based search for conference papers and articles was performed mainly from the scientific databases IEEE Xplore, Science Direct and ACM Digital as well as from Google Scholar. For search keywords, we used the following query: “Internet of Things” and either “greenhouse” or “livestock” or “aquaculture“ or “agriculture“. Then, we filtered out papers referring to IoT but not applied to the protected agricultural domain. Finally, we checked these papers selected from the previous step and picked out the final papers according to the problem they addressed, the solution proposed, impact achieved (if measurable), tools, systems and algorithms used.

The rest of the paper is organized as follows. Section 2 presents the structure of IoT in protected agriculture. The key technologies of IoT in protected agriculture are introduced in Section 3. The three important applications in protected agriculture are summed up in Section 4. We further discussed the challenges to be faced and the problems that need to be solved for IoT in protected agriculture as well as a few future development directions in Section 5. Finally, Section 6 outlines the conclusions of this paper.

## 2. The Structure of IoT in Protected Agriculture

### 2.1. Simple Review of Previous IoT Structure

At present, there is no unified opinion on the structure of IoT. The composition of IoT can be divided into three parts: object end, cloud end and network end. Smart items, smart devices and local intelligent systems interact directly with the physical world to form the “object end” of IoT (also known as “front end”). The cloud computing platform that provides computing, storage and other resources performs fusion processing and intelligent analysis of the sensing data from the object end and performs execution control to constitute the “cloud end” (also referred to as “back end”) of IoT. The communication infrastructure of the connection object end and the cloud end constitutes the “network end” of IoT. According to the different functions implemented by physical entity services in the IoT, IoT architecture established by the current service-oriented approach can be divided into the following two categories: cloud-centric IoT architecture and object-centric IoT architecture. In the cloud-centric IoT application system, the physical entity service implements basic physical information collection, local information processing and transmission of information to the cloud, but does not provide a directly accessible service interface. Mass storage, relational processing and knowledge mining of physical information are all implemented in the cloud. One of their typical representatives is uID IoT architecture, which is an IoT architecture developed by the uID center of the nonprofit standardization organization initiated by the university of Tokyo, Japan. As shown in Figure 1, its architecture consisted of ucode tags such as RFID, bar codes, RF and IR beacons, user terminals, a ucode resolution server and application information servers [16]. For software developers, the main challenge in building such systems is how to build a back-end cloud service platform that supports massive item information management and data processing. The cloud platforms that have been established so far mainly include Xively and Aneka [17].

In recent years, under the impetus of organizations such as Internet Protocol for Smart Objects (IPSO) Alliance, resource-constrained intelligent goods and sensing nodes can access the internet through the IP protocol and become devices that can be directly addressed in the network space [18]. Based on the service-oriented design method, researchers have proposed a variety of software architecture reference models for building IoT systems. M2M was a standard IoT architecture being developed by the European Telecommunications Standards Institute (ETSI) on the communication between machines and machines, which was designed for non-intelligent terminal devices to communicate with other intelligent terminal devices or systems through the mobile communication network. The functional architecture of M2M is shown in Figure 2: the M2M service capability layer (SCL) was deployed in devices, gateways and network domains with storage modules; the applications in devices and gateways accessed SCL through the dIa interface; the applications in the network domain accessed SCL through the mIa interface; the device or gateway interacted with the SCL in the network domain by the mId interface [19].

The 3-Tiers architecture combined the back-end cloud service with the front-end physical entity service and proposed a 3-layer information physical fusion system software architecture consisting of the environment layer, the service layer and the control layer. As shown in Figure 3, the physical components in the environment layer were services provided by the perceptron and the executor. The cloud service components in the service layer were services provided by platforms such as traditional cloud computing. As well, the following functions were implemented in the control layer: monitoring physical components and services, ensuring dynamic composability, ensuring service adaptability; and ensuring autonomous Cyber-Physical System management [20]. This architecture provides services for collecting physical data and interacting with the physical world. However, these services do not constitute an independent executable IoT systembut needto be called by the user or combined with the cloud service to complete the execution of related tasks.

IoT-A was an IoT software architecture that implemented horizontal interconnection and interoperability between local IoT systems supported by the EU’s 7th Framework Project. Its reference model is shown in Figure 4. The local object association system with different sensing and communication technologies was abstracted into an IoT resource model that provides unified services and the components were divided into several components according to their roles, function granularity and abstraction level, including device component, communication component, service organisation component, IoT process management component, virtual entity component, IoT services component, security component and management component [21]. and. The IoT-A architecture establishes virtual entity services on the resource model and provides a higher level of abstract interface for the construction of the IoT application system through various modules. The IoT-A architecture has high flexibility and wide applicability as it meets application needs in the form of business processes.

### 2.2. Structure of IoT in Protected Agriculture

Based on the actual situation of protected agriculture and the experience of other scholars, we proposed a five-layer IoT architecture [22]. As shown in Figure 5, these layers are briefly described below:
(1).Perception layer: This layer consists of various sensors, terminal devices, agricultural machinery, wireless sensor network (WSN), RFID tags andreaders, etc. The common sensors are environmental sensors, animal and plant life information sensors and other sensors related to agriculture. Through these sensors, information such as temperature, humidity, wind speed, plant diseases, insect pests and animal vital signs can be obtained. The collected information is simply processed by the embedded device and uploaded to a higher layer through the network layer for further processing and analysis.(2).Network layer: The network layer is the infrastructure of IoT, which includes a converged network formed by various communication networks and the internet. The transmission medium can be wired technology such as CAN bus and RS485 bus or wireless technology like Zigbee, Bluetooth, LoRa and NB-IoT. The network layer not only transmits various kinds of agricultural related information collected by the perception layer to the higher layer, but also sends the control commands of the application layer to the perception layer, so that the related devices of the sensing layer take corresponding actions.(3).Middleware layer: IoT can provide different types of services for different devices. The technical specifications (processor, power supply, communication module) and system of each device are different and different devices cannot be connected and communicated with each other, which leads to heterogeneity problems. The middleware layer aggregates, filters and processes received data from IoT devices, which greatly reduces the processing time and cost of the above issues and provides developers with a more versatile tool to build their applications. It also simplifies the steps of new service development and new device deployment which enables them to integrate more quickly into older architectures, improving the interoperability of IoT.(4).Common platform layer: The common platform layer is responsible for the storage, decision-making, summary and statistics of agricultural information and the establishment of various algorithms and models for agricultural production process such as intelligent control, intelligent decision making, diagnostic reasoning, early warning and prediction. This layer is composed of cloud computing, fog computing, edge computing, Big Data, machine learning algorithm, other common core processing technologies as well as its establishment model.(5).Application layer: The application layer is the highest level of the architecture and the place where IoT’s value and utility are most apparent. There are lots of intelligent platforms or systems in this layer for the environmental monitoring and control of plants and animals, the early warning and management of diseases and insect pests, and agricultural product safety traceability, which can improve production efficiency as well as save time and cost.

## 3. The Crucial Technologies of IoT in Protected Agriculture

### 3.1. Sensor Technology

Sensor plays an indispensable role in obtaining environment, plant and animal information and it is also one of the technical bottlenecks in the development of IoT. In recent years, sensor technology has rapidly developed with the emergence of new materials and methods [23]. The conventional environment sensors such as temperature, humidity, light intensity, heat and gas sensors have been relatively mature and widely deployed in protected agriculture. Nowaday, about 6000 research and production organizations engage in sensor research from more than 40 countries, including famous manufacturers such as Honeywell, Foxboro, ENDEVCO, Bell & Howell and Solartron companies. There are three main types of widely used and researched agricultural sensors: physical property type sensors, biosensors and micro electro mechanical system sensors (MEMS). The physical property type sensor realizes the signal conversion through the physical change of the sensitivity of the material of the sensor itself, which is mainly temperature, humidity and gas sensor. The biosensor [24] uses the organism itself as a sensitive component to transmit information according to the reaction of the organism to the outside world and includes enzyme sensor [25], microbial sensor, adaptive sensor [26], etc., which is mainly used to detect pesticide residue, heavy metal ion, antibiotic residue and toxic gas [27,28]. The MEMS sensor is the embodiment of the new generation of research and development technology in the field of sensors with small size, low power consumption, low cost and reliability [29]. 

Although sensor technology has made great strides in both principles and manufacturing processes, there are still some open issues. A typical example among those open issue is the research and development of soil sensors and vital sign information sensors of animal and plant, which is an urgent conundrum to overcome in long time. The complex composition as well as different physical and chemical properties of soil turn the rapid and situ measurement of soil nitrogen and other elements into an international difficulty. Since the life process of animals and plants is very complicated, in order to realize the accurate detection of information about their life, it is necessary to make breakthroughs in the physical mechanism and measurement model of the life process. Related research has been carried out using physicochemical properties, spectroscopy, multispectral, hyper-spectral, nuclear magnetic resonance as well as other detection methods [30,31,32,33]. 

Another problem is the promotion and application of these sensors. For one thing there is still a lack of uniform technical standards in the collection, transmission and the interface with the platform and human-computer interaction interface; for another, the cost has limited the large scale application of fresh sensors in intelligent agriculture. Currently, the cost of sensing equipment for the collection and transmission of animal and plant status is prohibitively expensive for ordinary peasants. The number of nodes in a sensor network can often reach tens of thousands. To make the sensor network practical, the cost of each node must be controlled within $1, but now the price is as high as $80 [34].

### 3.2. Data Transmission Technology

In this section, the details of data transmission technology are discussed, which is one of the vital technologies for real-time dynamic acquisition of agricultural information. Different from industrial application, agricultural information acquisition is not demanding in timelines and a slight delay in the transmission process has little effect on agricultural production. 

Compared with wired transmission technology such as fieldbus, wireless communication technology has advantages of low construction and maintenance cost, low power consumption and excellent extensibility. Recently, most researchers, organizations and manufacturers choose it to build up their WSN in environment monitoring [35,36], automatic irrigation [37] and remote control [38]. As shown in Table 1, the summary of popular wireless technologies and basic parameters was presented. Different vendors and research institutes have developed their own wireless devices based on these wireless protocols, which has somewhat increased the heterogeneity of IoT. In addition, wireless signals of different protocols located in the same band can cause mutual interference such as ZigBee, Wi-Fi and Bluetooth [39]. Wi-Fi has high communication speed but high power consumption, which is suitable for sensor network deployment at fixed points. Although Bluetooth has high security, its communication distance is too short and power consumption is high, which is suitable for short-time close-range networking. ZigBee has the advantages of low consumption, low cost and self-organization and each node can be used as a relay station for transmitting data of adjacent nodes. So it easily expands the coverage of the sensor network and is an ideal long-distance, large-range sensor networking.

In recent years, the Low Power Wide Area Network (LPWAN) is one of IoT hotspots and thought as a novel technology with great development potential, which is attributed to its huge advantages such as ultra-long distance communication and it can be used as an important complement of the traditional wireless local area network (WLAN) and mobile cellular mobile communication technologies (such as GSM and GPRS). At this stage, LPWAN has many technical standards, which can be classified into two categories. The first is a proprietary patented technology that works under an unlicensed spectrum, such as LoRa, Sigfox and RPMA. The second is a cellular communication technology that works under a licensed spectrum, such as EC-GSM, eMTC and NB-IoT. Due to the use of dedicated frequency bands and unified deployment by operators, licensed spectrum technology has carrier-class security and low interference characteristics, which forms full network coverage and operation [40,41]. The unlicensed spectrum uses the common spectrum without paying high spectrum costs but requires special handling of co-channel interference. There are many benefits of IoT deployments based on LPWAN in the agricultural environment. Firstly, the maximum connection distance of the LPWAN terminal device to the base station varies from several kilometers to several tens of kilometers, which saves the trouble of setting up and maintaining routing devices. Compared with the 2.4 GHz band, LPWAN adopts the sub-1GHz band to acquire less attenuation and multipath fading caused by obstacles and dense surfaces like concrete walls [42,43]. Moverover, the sensitivity of state of the art LPWAN receivers reaches as low as –130 dBm due to modulation techniques such as narrowband and spread spectrum. Secondly, ultra low power operation is another advantage of LPWAN which is achieved by duty cycling mechanism, lightweight medium access control protocols and offloading complex tasks [44]. Thirdly, most LPWAN technologies support a large number of device connections by diversity techniques, adaptive channel selection and data rate technologies, etc. [45]. Finally, the LPWAN connects a large number of end devices, while keeping the cost of hardware below five dollars [46]. LPWAN technologies target a diverse set of applications with varying requirements in protected agriculture and therefore it should provide some sort of quality of service (QoS) over the same underlying LPWAN technology. However, current LPWAN technologies provide no or limited QoS.

Meanwhile, mobile cellular communication technology has undergone four generations of technology updates and the fifth-generation mobile communication technology (5G) was announced in 2016, which will make large file transfer such as agricultural images and audio a reality and bring new impetus to the agricultural IoT. From the perspective of future research, the research of wireless sensor networks focuses mainly on communication, energy conservation, network control and its application in agricultural monitoring.

### 3.3. WSN

The WSN consists of a lot of sensor nodes that are usually powered by battery and it is a multi-hop self-organizing network system formed by wireless communication to collaboratively sense, collect and process various information of the perceived object in the network coverage area [47,48]. It can be divided into terrestrial WSN and wireless underground sensor networks (WUSN). The agricultural sensors are usually planted into soil and lower frequency wireless technologies are more selected due to low attenuation in WUSN. Besides, antenna size and energy consumption in WUSN is higher than terrestrial WSN [49]. With development of LPWAN, IoT may not need a mesh-style WSN with power-based routing, where one node forwards packets of other nodes. 

### 3.4. Cloud Computing

Cloud computing grew out of distributed computing, parallel processing and grid computing. It is an Internet-based computing system that provides hardware services, infrastructure services, platform services, software services and storage services to various IoT applications. Its essence is a system for dynamically deploying and allocating/redistributing as well as dynamic monitoring of virtualized computing and storage resource pools, thereby providing users with computing, data storage and platform services that meet QoS requirements [50,51]. It played a strong role in promoting the development of agricultural IoT. Firstly, cloud computing can provide farmers with cheap data storage services for text, image, video and other agricultural information, which greatly reduces the storage costs of agricultural enterprises [52]. Secondly, it is difficult to make direct use of these raw data for making decisions based on the technical level of farmers. Agricultural experts can only make accurate judgments and give suggestions based on quantitative analysis. Only cloud computing can support intelligent large-scale computing systems [53,54]. Thirdly, cloud computing could provide a secure platform for the development of various IoT applications such as crop monitoring [55].

### 3.5. Edge Computing

Edge computing refers to a new computing model that performs calculations at the edge of the network [56]. In the edge computing, the downlink data of the edge represents the cloud service, the uplink data represents IoT service and the edge of edge computing refers to any computing and network resources between the data source and the cloud computing center path. Because the computing task is partially migrated to the network edge device, edge hormone can improve data transmission performance, ensure real-time processing and reduce the computational load of the cloud computing center. Moreover, edge computing better protects data because processing occurs closer to the source than in cloud computing [57,58].

### 3.6. Machine Learning

Machine learning (ML) is a smart way for computers to simulate people’s learning activities, acquire new knowledge, continuously improve performance and implement their own perfection. In recent years, machine learning has achieved great success in algorithm, theory and application [59] and have been combined with other agricultural technologies to maximized crop yield while minimizing the input costs [60]. The main machine learning algorithms are naïve bayes, discriminant analysis, K-nearest neighbor, support vector machine (SVM) [61], K-means clusterin, fuzzy clustering, gaussian mixture models, artificial neural network (ANN), deep learning [62], decision tree algorithm and so on [63]. ML can discover the internal connection between messy, modelless and complex agricultural data, make accurate predictions and provide a theoretical basis for agricultural decision-making. Machinelearning technology is playing an important role in crop breeding, disease identification, pest and disease prediction, intelligent irrigation planning and agricultural expert systems [64,65,66,67]. For example, ML technology can analyze past farmland data, including the performance of crops under different climatic conditions and the inheritance of a particular phenotype. As well, ML technology can explore the association rules and then build a probability model to predict which genes are most likely to participate in the expression of a certain good trait of the plant, which can help the breeding expert conduct a reasonable breeding experiment [68,69,70]. Kyosuke et al. came up with a method for detecting the maturity of a single intact tomato-based ML, which consisted of three steps: pixel-based segmentation, blob-based segmentation and individual fruit detection. In the first two steps, decision trees were generated based on features such as color, shape, texture and size and image segmentations were conducted using the generated trees. Finally, the automatical detection of individual fruit in multiple tomatoes was achieved by X-means clustering method. The results of the tomato detection image test showed that their method achieved a recall of 0.80, while the precision was 0.88 [71]. Rahnemoonfar et al. proposed deep learning architecture for counting fruits based on convolutional neural networks. They used synthetic images to train the neural network to save time-consuming collect and annotate. The network was trained for three epochs on 24,000 synthetic images using an Adam optimizer and 100 randomly chosen images were tested. The experimental results demonstrated that their method had higher average accuracy as comparing with others [72].

### 3.7. Big Data

The protected agriculture produces billions of dynamic, complex and spatial data including soil databases, greenhouse environment data, livestock vaccination records and government investment data. Compared with relational data structures that are logically expressed using two-dimensional tables, agricultural data is more unstructured and has a large number of hypermedia elements such as expert experience, knowledge and agricultural models in the form of text, charts, pictures, animations and voice/video [73]. The “big” of these data can be reflected in four dimensions: volume, velocity, variety and veracity [74]. Big Data technology is able to find out the internal links of collected data through information mining and other means, discover new information and provide data support for the next operation. The most commonly used techniques to deal with Big Data technology are image processing, modeling and simulation, machine learning, statistical analysis and geographical information systems (GIS) [75].

## 4. IoT Applications in Protected Agriculture

### 4.1. Plant Management

Compared with open-field agriculture, protected agriculture provides a relatively suitable and controllable environment for crop growth by greenhouse technology, which to some extent is free from the constraints of the natural environment and promoted the intensive and efficient use of agricultural resources. However, spatial and temporal variability of crops’ growth environmental parameters in the protected agriculture are strong and affect each other [76,77], which made it difficult to adapt to the growth of different types of plants in different growth stages by traditional cultivation and environmental regulation. Therefore, it needs higher accuracy in the aspect of monitoring and control. Many works have attempted to design and test types of monitoring and control systems to adjust greenhouse environmental parameters such as air temperature and humidity, light intensity and CO_2_ concentration based on IoT and their results have proven it is technically and economically feasible [78]. At the low level of IoT development, the environmental data were simply processed and usually presented in sheet and plot form [79,80]. Later, some studies collected huge amounts of data to set up various models based on plant growth or greenhouse climate [81,82,83], which contributed to predicting the crop yield and environment parameter changes to help farmers better manage greenhouse. CAI et al. developed a low-cost smart greenhouse remote monitor system based on IoT. The system used STM32F103 single chip machine and sensors to collect environmental information and transferred the data to cloud by LPWAN. Security link was established with TLS, which reduced system costs while ensuring the security of data transmission. At the same time, they adopted a time series database to store data so that tremendous storage space can be saved [84]. He et al. have reported a greenhouse temperature intelligent control system based on NB-IoT. The relative error of greenhouse environmental information collection was below 1%, the average control accuracy was 3.57% (±1.0 °C), the transmission distance was not limited and the automatic regulation of crop growth temperature was realized [85]. 

Nowaday, with the development of cloud computing, ML, etc., IoT solution can easily achieve smart data processing and analyzing at low cost and in a convenient way [86,87]. At the same time, greenhouse technology has also undergone several generations of upgrades and has now evolved into highly mechanized and automated plant factories. Deng et al. have implemented a closed-loop control system in a salad-cultivating plant factory based on the kinetic model. Both digital simulation and real-time results demonstrated the system can work stably under the internal variations and external disturbances [88]. Zamora-Izquierdo et al. developed a smart farming IoT platform based on edge and cloud computing, which was designed for soilless culture greenhouses at low cost. The platform was composed of local, edge and cloud parts: the local part dealt with data gathering and automatic control though Cyber-Physical Systems, the edge part took main management tasks and could improve the stability of this systems, the cloud part was in charge of data analytics. Compared with a regular open control, the platform saved more than 30% water [89]. Katsoulas et al. have designed an online irrigation system for hydroponic greenhouse crops and their results indicated it increased water and fertilizer use efficiency by 100% [90]. Liao et al. implemented a IoT-based system in an orchid greenhouse to monitor the environmental factors and the growth status of phalaenopsis. They developed an image processing algorithm to estimate the leaf area of phalaenopsisthe and identified the relationship between growth of plant leaves and environmental factors. The proposed system could provide quantitative information with high spatiotemporal resolution for floral farmers and contribute to updating farming strategies for phalaenopsis in the future [91].

Diseases and insect pests bring a great threat to the growth of crops as and traditional technology and chemical prevention and control has a certain lag and negative impact [92,93]. The rise of IoT has brought more efficient and smarter solutions to crop disease and pest control. Many types of IoT sensors can collect information about location, greenhouse environment status, crop growth and pest situation anywhere in real-time, which helps farmers keep track of crop pests and diseases. Then, all raw image and data are sent to cloud centers and later processed and analyzed by various models and algorithms based on different diseases and pests [94,95]. Finally, these cloud centers generally provide farmers with the following services: disease or pest identification, disaster prediction and warning and recommended governance measures from expert systems. Tirelli et al. proposed a pest insect trap automatic monitoring system using ZigBee technology, which can estimate the insect density by collected data from different sites and send a warning message to farmers when it exceeded the set value [96]. Ahouandjinou et al. proposed a pest monitoring system which detected the presence of pests by the acquisition of ultrasound and assisted others in building up protocols for early exterminate of the pests [97]. Foughali et al. presented a potato late blight prevention and decision support system using cloud IoT and helped the agriculturists take effective action to treat this disease [98]. Both Semios and Spensa Company launched their own integrated pest management system, which was able to count the number of pests by images, as well as characterize and capture the insects. At present, pest and disease warning research mainly provides medium and long-term warning based on historical data, which can provide macro guidance for crop production but low timeliness. Therefore, future research should focus on online monitoring as well as diagnosis and early warning of agricultural diseases.

### 4.2. Animal Farming

Livestock and aquatic product farming is an important part of protected agriculture and an area where IoT applications have achieved good results. To achieve good control effects in animal breeding, IoT should not only overcome harsh environmental factors, but also pay attention to the effects of animal behaviors [99,100]. IoT have been applied in monitoring and management of environment, animal, feed and farming process [101,102]. The livestock monitoring items include information such as body temperature, weight, behavior, exercise volume, food intake, disease information and environmental factors, which can help people understand animal’s own physiological and nutritional status and adaptability to external environmental conditions. In aquaculture, management projects focus on water quality such as dissolved oxygen content, water temperature and pH value [103,104] because water quality greatly affects the growth of aquatic animals. With animal growth and nutrient optimization model and intelligent IoT equipment, it is possible to realiaze automatic feeding and optimal control of feeding time and intake according to their growth cycle, individual quality, feeding cycle and eating situation [105,106]. The Osborne Industrial Company has produced TEAM automated electronic sow feeding (ESF) stations including pregnancy stations and estrus detection stations. The ESF stations identify the sow through the RFID tags worn by each sow and deliver the corresponding quantity and type of feed based on information such as parity, lyrical condition and gestational age. The estrus detection stations can detect rutting sows in the sow population and their detection accuracy is 7% higher than the manual ones. Encinas et al. presented a distributed monitoring IoT system for water quality monitoring. Their model was able to help fisheries acquire water quality parameters such as pH and temperature data in real time to optimize pond resources and prevent unwanted conditions [107]. Soto-Zarazu´a et al. proposed an automated recirculation tilapia farming system based on the fuzzy algorithm. The results of this work showed that it saved water by 97.42% and the water quality environmental parameters were controlled within the acceptable range of tilapia culture [108]. Many researches have focused on analysis of animal behavior, health care and diagnosis and warning of diseases based on IoT [109,110,111]. Yazdanbakhsh et al. proposed an intelligent livestock surveillance system. They attempted many machine learning algorithms to process raw data of healthy and ill cows and finally obtained good results used a wavelet-domain ensemble classifier with 80.8% sensitivity and 80% specificity [112]. Wens Group, the largest livestock breeding enterprise in China, took the lead in carrying out research on the animal husbandry based on IoT and built the corresponding system for monitoring livestock vital signs, behavior and breeding environment information [113]. Liu et al. collected a variety of penaeus vannamei information such as real-time data on the culture environment, disease image data and expert disease diagnosis and treatment and finally, constructed a remote intelligent diagnosis model based on IoT [114]. Liu et al. proposed a web-based combined nutritional model for precisely predicting the growth, feed amount and nitrogen phosphorus output of gibel carp [115]. The current online diagnostics and early warning of animal diseases is at an early stage due to the lack of corresponding sensors and models. With upgrading of hardware and artificial intelligence algorithms, future research should focus on the improvement of Big Data depth algorithm and the establishment of comprehensive models of animal behavior and disease. 

A few studies have made in-depth research in monitoring animal odor and hazardous gas produced during the breeding process such as ammonia gas and sulfur dioxide [116]. Pan developed an electronic nose network system for monitoring and real-time analysis of odors from livestock farms. They placed e-noses in and around the farm to build up a wireless network to measure odor compounds as well as environmental parameters [117]. 

### 4.3. Agri-food Supply Chain Traceability

Nowday, agricultural products/food safety issues are receiving worldwide attention and their safety traceability is one solution accepted by all parties of Agri-food domains. Governments in many countries and regions have promulgated laws and regulations to promote the establishment of food traceability system and strengthen the supervision of agricultural products/food safety. The Agri-food supply chain traceability IoT-based system can ensure food safety and quality at each link of the production, from the cropland to the consumer (Figure 6), which could help consumers establish confidence in food safety and contribute to sustainable development of the whole food industry [118]. 

In past years, a lot of countries have established the traceability platform for meat, milk, fish and agri-food products based on IoT [119,120,121]. However, the fresh food cold chain logistics traceability has also drawn widespread attention [122,123]. RFID technology still played an important role in Agri-food supply chain traceability due to its small size and low cost [124]. As a novel technology, near field communication (NFC) has been progressively developed and used because of its safe and simple operation [125,126]. A common problem in the development of IoT is its asynchronous heterogeneous data flow and distributed features. This requires the traceability system to establish uniform and accurate identification naming rules to facilitate quick and unique retrieval of information on a farm product [127]. As the deployment of IoT infrastructure is completed, the supply chain will move towards virtualization which is no longer required physical contact. The virtualization of an agri-food supply chain helps administrators better monitor, control, plan and optimize food supply chain processes [128]. Wang et al. proposed a food safety pre-warning system based on association rules mining and IoT. It included four modules: information source, warning analysis, reaction and emergency feedback. First, they accessed food critical information through data processing and analysis of food traceability data for safety assessment. On this basis of this data, the system used the associated data mining method to discover the connection between them and then obtained the food safety risk analysis results. Subsequently, the system made a corresponding early warning response based on this result [129].

Recently, the increasing incidents of food falsification have not only brought about great economic losses, but also undermined consumers’ confidence in food safety. The blockchain technology with decentralization, non-tamperability, development transparency and machine autonomy features brings new solutions to the above problems [130,131]. Tian et al. reported a new decentralized traceability system based on IoT and blockchain technology, which can provid an open, transparent, neutral, safe and reliable information platform for producers, governments, consumers and other stakeholders [132]. Kaijun et al. combined the supply chain and blockchain technology to propose a double chain architecture agricultural traceability system by studying the dual chain structure and its storage mode, resource rent-seeking and matching mechanism and consensus algorithm [133].

Now, with the innovation and maturity of related technologies, it is not difficult for researchers to develop a complete set of traceability systems. The rise of artificial intelligence technology enables existing traceability systems to provide automation, intelligence and human services to businesses and consumers. Chen et al. adopted fuzzy cognitive maps and fuzzy rule method to come up with an autonomous tracing system for backward design in food supply chain based on IoT, which better traced food product problems [134]. However, there are still some problems in food safety traceability. It is worth noting that the current agricultural product safety traceability system only focuses on a certain level of supply chain or a certain type of goods. With the participation of multiple projects or multiple sessions, we believe that the focus of future research should be on more complex and systematic supply chains. 

## 5. Disscussion

### 5.1. Hardware and Software Challenges

When IoT technology is applying in protected agriculture, it is inevitable to encounter challenges from all aspects. In the perception layer, various devices must face the harsh and complex greenhouse environment. Strong solar radiation, high temperatures, high humidity, strong vibrations and other hazards can easily damage or destroy sensors or end devices. Moreover, the daily activities of livestock can interfere with the work of the sensor or execution nodes, resulting in poor detection and control. In a general way, the information collection nodes rely on the battery with limited power to maintain its work because the behavior of frequent battery replacement consume a lot of energies and money. Therefore, it is an urgent need to make great progress in developing low-power acquisition equipment, energy-saving sink node routing protocols and energy-balanced communication algorithms [135].

The devices places in various places generate incalculable data and the storage of such data is also a huge challenge for some small servers. Those agricultural IoT data have real-time, dynamic, granular and fragmented characteristics, which brings great challenges to intermediate design, data large-scale screening, screening, mining, processing and decision analysis techniques. Moreover, existing databases cannot store unstructured data such as audio, video and images. As a result, quite a lot real-time sensing data are not fully exploited. Besides, some models and algorithms are not enough to to reflect objective reality, so that they cannot effectively guide agricultural production. 

### 5.2. Network Challenge

In protected agriculture, the sensor and device need to continuously operate under the harsh and varied environment and their arrangement is sparse and irregular. Compared with the wired network with high wiring costs, the wireless network has the advantages of low cost, good networking flexibility and high scalability and therefore becomes the main application mode of the current monitoring system. Many actual deployments have displayed that common wireless communication technologies are affected by temperature, humidity andbuildings or other spatial barriers [136,137,138]. The growth of animals and plants changes also result in background noise [139] because of the multi-path propagation effects, which lows reliability of data transmission. It is easy to isolate a node in the original communication network due to low node deployment density and the above obstacles in protected agriculture. If this node is responsible for communication routing tasks of many devices and this node is damaged and isolated from the network, this will cause partial or even paralysis of most of the network. Most relevant theories and research needes on the effects of the crop growth, the thickness and material of wall, and the radio signal transmission loss and electromagnetic wave transmission between soil and air interface, etc. To determine the optimal position, height and network topology of sensor nodes under different environmental conditions. On the other hand, ad hoc intelligent wireless network technology is needed to reduce or avoid paralysis of large-area networks. The rise of LPWAN may solve e above paralysis problems of WLAN due to its long communication distance, which obviates the need for the dense and expensive deployments of relays and gateways altogether. However, the cost of erecting LPWAN base stations is high.

The process of recording various agricultural products’ status from the field to the table generates a large amount of data, which brings great challenges to wireless communication with a low transmission rate. At same times, it also brought network nodes many problems in energy distribution, data calculation, storage and communication. Therefore, there is an urgent need to develop high-capacity data and real-time broadband communication standards for technologies in complex agricultural application environments.

### 5.3. Security Challenge

The problems of security and privacy are thought as crucial challenges in applications of protected agriculture due to real-world examples of losses due to vulnerabilities, network attacks or privacy issues. As a multi-network heterogeneous converged network, it not only has the same security problems as sensor networks, mobile communication networks and the Interne, but also has its particular issues such as privacy protection issues, authentication and access control issues for heterogeneous networks, storage and management of information issues, etc. 

In the perception layer, security threats are mainly against the WSN and RFID. In agricultural applications, many of the sensor nodes are deployed in the long-term unsupervised farmland environment and the wireless network itself is open, so the wireless transmission is vulnerable to external interference and illegal users. Coupled with the fact that these devices are respectively placed elsewhere, a single security measure is not enough [140]. The main threats to WSN include eavesdropping attack, node capture, replay attack and information wiretapping. So multiple security mechanisms such as anonymity mechanism, data encryption and identity authentication mechanisms are necessary [141,142].

The security threat of RFID refers mainly to the attacks that are encountered when the RFID reader and the tag communicate and such attacks usually lead to user privacy. In addition, because the RFID tag is rewritable, the security and validity of the RFID tag data will not be guaranteed. There are lots of ways to enhance RFID tag security such as modifying tag frequency, enciphered data. Sensor nodes are different from RFID tag because they are active and relate to the dynamic properties of things. Therefore, when applying encryption algorithms, key distribution policies, intrusion detection mechanisms and security routing policies in specific devices, we need to consider the limitations of their own hardware.

Since networks of different architectures need to be connected to each other, there are even greater challenges in terms of security authentication across network architectures. The transport layer will encounter the following security challenges: proxy attacks, DoS (Denial-of-Service) attacks, malicious code injection, man-in-the-middle attacks, attacks across heterogeneous networks, etc. The common measures dealt with these network layer threats and attacks are: authorization, authentication, encryption, anti-virus protection, etc. IoT terminal devices in protected agriculture have the characteristics of weak computing power and small storage space. Due to limited resources, IoT encryption technology must be an easy to implement, lightweight security technology that is suitable for use in sensitive information environments [143,144].

When massive agricultural data is transmitted to the application layer, in addition to the intelligent processing of data, data security and privacy should also be considered. At this level, a large amount of private data about users is collected, such as the consumption of agricultural materials, the purchase information of fruits and vegetables and the location of agricultural machinery. Therefore, the database access control strategy should be strengthened on the basis of intelligent data processing. When different users access the same data, they should limit their permissions and operations according to their security level or identity, effectively ensuring data security and privacy. On the other hand, application layer service providers should accelerate the development and upgrade of authentication mechanisms and encryption mechanisms for application scenarios such as agricultural production, processing, transportation and consumption.

### 5.4. Other Challenges

There are other challenges that need to be addressed, which are related to the further application of IoT technology in protected agriculture. Thousands of devices which have huge differences in processor, memory, communication protocol and programming language are deployed in protected agriculture, which inevitably results in heterogeneity issues. Besides, most of devices in protected agriculture are connected to the cloud or others by the non-standard heterogeneous interfaces [14,22]. This has led to problems of both device heterogeneity and data heterogeneity. The device heterogeneity problem affects the scalability of IoT in protected agriculture and the data heterogeneity problem hinders the use of fusion information by models [145]. Although many research institutes, hardware and software manufacturers and related organizations have made a lot of contributions to the standardization and deployment of IoT in agriculture, it still needs to establish a complete and clear structure, protocols and standards to connect various heterogeneous devices and services [146,147]. 

The cost has always been a barrier to the large-scale application of IoT to ordinary farmers, especially in developing countries like China and India. Despite a large number of embedded devices and platforms have emerged and the cost of hardware and software is rapidly decreasing, but here is another situation in high-quality and high-precision sensors and devices. Moreover, some funds are needed to train farmers to become proficient in the use of IoT equipment. In order to be able toplay a better role in the future, the cost of sensing, transmission, analysis and application components and equipment on the IoT need to further reduce in order to lower the price and make it more affordable in the next decade [148]. Government departments should recognize the significant potential for the development of IoT in protected agriculture and increase investment and support through policies and funds to attract more capital and achieve rapid development and application.

In addition, the usability of the terminal device needs to be further improved, for example, the user interface needs to be friendlier andthe device needs to be more convenient in size and easy to carry. 

IoT brings many benefits to ecological environment and brings some problems.. More and more equipment is being placed in agriculture andso the issues of environmental friendliness should receive some attention in the future. As the e-waste generated by IoT devices and the consumption of energy are gradually increasing, the sustainable development of the ecological environment is also one common concern. Examples of interesting research topics include the use of degradable materials and renewable energy sources, the reduction of equipment size and the use of energy-saving algorithms and new green ICT technology [149,150].

The application of the Internet of Things in facility agriculture needs to consider the socio-economic consequences. On the one hand, it increases the level of intelligence in agricultural monitoring, control and decision-making, thereby increasing the efficiency and quality of agricultural production. At the same time, it will also change the demand for the existing agricultural workforce, causing some people who are engaged in simple repetitive tasks to lose their jobs, which will cause a series of social problems.

### 5.5. Future Prospects

IoT technology is being applied to many fields of society, economy and life at an unprecedented speed, making human society step into a new era of high intelligence. IoT technology is evolving and mature, and many novel deployments and applications are constantly being built in protected agriculture. Based on the above analysis and discussion of the key technologies and applications of IoT in the protection of agriculture, we will present its future development prospects and key research and development directions in the following aspects. In the perception layer, the development of sensors should focus on new sensitive materials, mechanisms, processes and methodologies as well as low power consumption and low cost. Besides, we must accelerate the development of sensors and rapid detection devices soil, animal and plant life. At the network layer, we should focus on high-capacity data real-time broadband communication standards and technologies for complex agricultural application environments, high-reliability, adaptive, low-power network deployment and management strategies and algorithms. With wide coverage, high capacity, low power consumption and low cost, LPWAN is ideal for some agricultural scenarios where text-based, video and voice service support is not required and coverage, power consumption and cost are critical. Future research and deployment should focus on IoT solutions based on LPWAN. In the application layer, governments and organizations should develop unified standards for the sensors and identification interface devices, data transmission communication protocols, multi-source data fusion analysis processing and application services in protected agriculture through international cooperation or negotiation. Software developers should develop large open source databases and signal processing algorithm libraries for different areas of facility agriculture. For universities and research institutes, cloud-based agricultural intelligent decision-making models, multi-source data-based information fusion algorithms, agricultural Big Data mining technologies, distributed intelligent processing systems and lightweight IoT authentication, encryption and authorization mechanisms will be the key research directions for the future. With an increased presence of IoT technology in protected agriculture, its potential for the refined management of crop, livestock and aquatic animals will be recognized. Besides, as a technical means to monitor the production, processing, circulation and consumption of agricultural products, IoTwill play an increasingly important role in food safety.

## 6. Conclusions

In this survey, we present a comprehensive review of the state-of-the-art in IoT deployment for protected agriculture applications. First, a simple review of previous IoT structure was given. Secondly, the basic IoT architectures in protected agriculture were introduced. Next, discussions of sensor, data communication, cloud computing, edge computing, ML and other vital IoT technologies in protected agriculture are elaborated. Then, we highlight various IoT applications and their potential to solve various farming problems in protected agriculture. By classifying the research and deployment literature on IoT in protected agriculture, three important application fields were given: plant management, animal farming and agri-food supply chain traceability. Finally, a detailed analyses of IoT research challenges and future prospects were outlined.

The survey of the existing works directs us concluding remarks. The future prospects of IoT in protected agriculture are expectant, but the challenges mentioned above must handle. In order to cope with its complex and changing agricultural environment, the hardware devices must be fully upgraded to further enhance their universality, reliability, expansibility, endurance and intelligence level, while reducing costs and operative difficulty. Secondly, the local network must be protected from interference from other networks. In addition, the interoperability, filtering and semantic annotation of data generated by the various devices of IoT must be realized to a certain extent. Only in this way can we optimize the Big Data decision support model by using a large amount of heterogeneous data. Security, anonymity and control over the access rights on the information is vital for such a system to be adopted. Finally, the impact of IoT on the agroecological environment and social economy should be considered in the application of protected agriculture, to realize the sustainable development of agricultural environment as quick as possible.

## Figures and Tables

**Figure 1 sensors-19-01833-f001:**
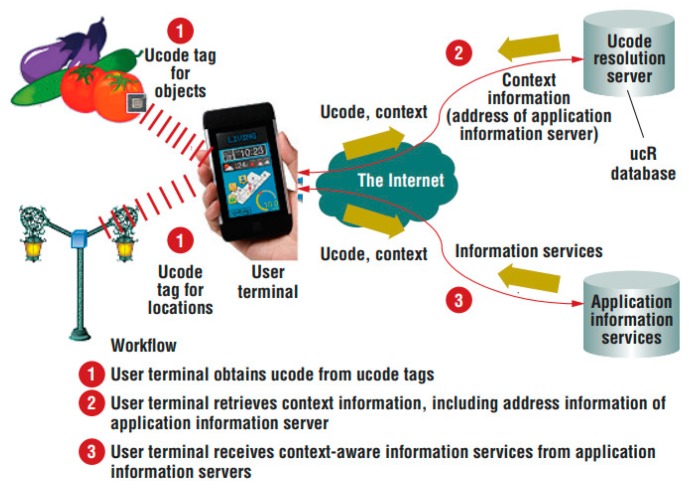
uID IoT architecture [16].

**Figure 2 sensors-19-01833-f002:**
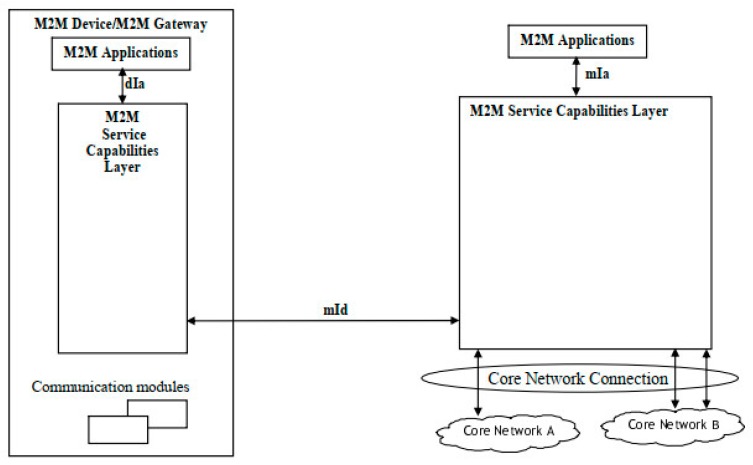
M2M IoT architecture [19].

**Figure 3 sensors-19-01833-f003:**
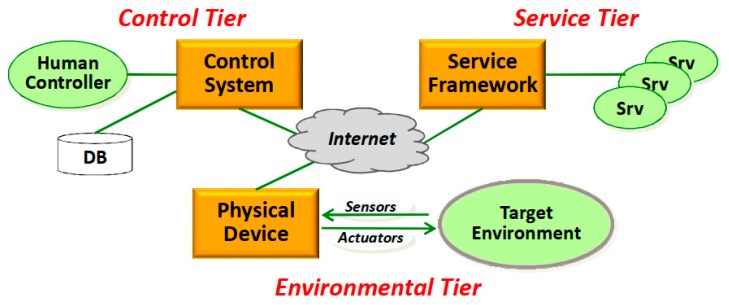
3-Tiers IoT framework [20].

**Figure 4 sensors-19-01833-f004:**
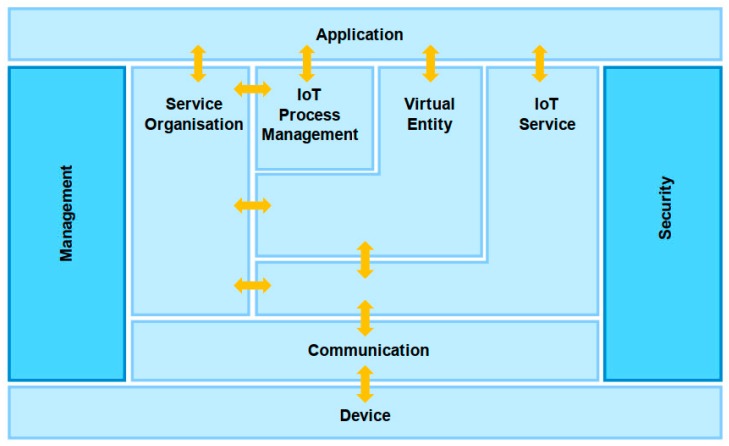
IoT-A IoT framework [21].

**Figure 5 sensors-19-01833-f005:**
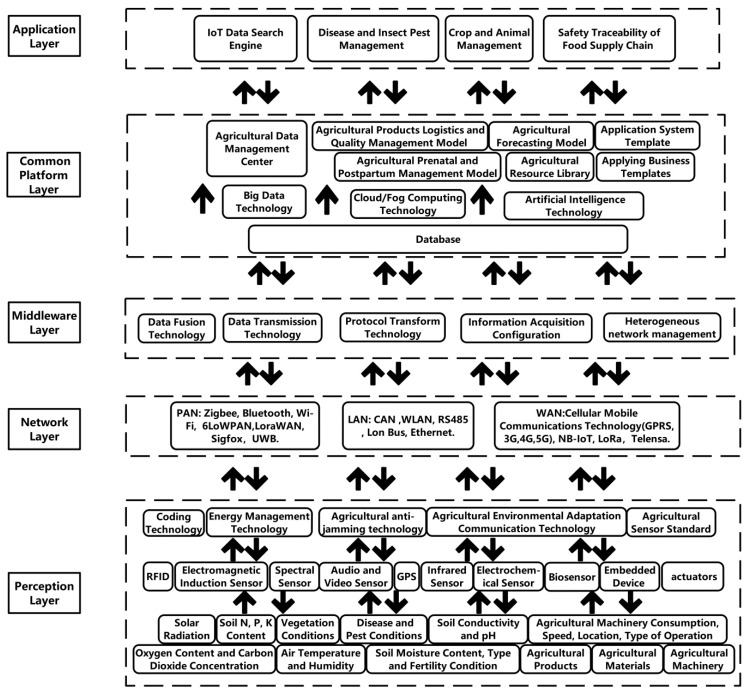
Structure of IoT in protected agriculture [22].

**Figure 6 sensors-19-01833-f006:**
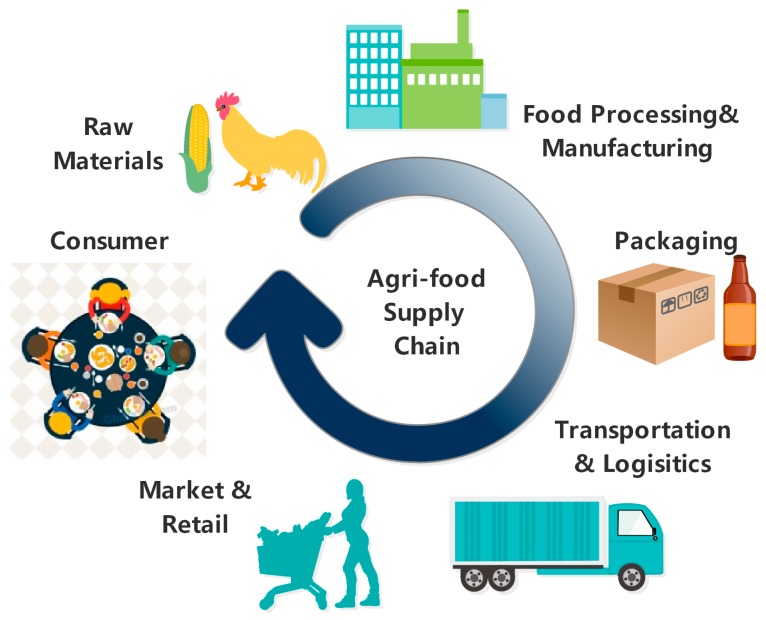
Schematic representation of the Agri-food supply chain from the grower to consumer.

**Table 1 sensors-19-01833-t001:** Summary of popular wireless technologies and basic parameters.

Wireless Technology	Wireless Standard	Frequency Band	Network Type	Transmission Range	Data Rate &Power
Wi-Fi	IEEE802.11 a/c/b/d/g/n	2.4 GHz, 5–60 GHz	WLAN	20–100 m	1 Mb/s–6.75 Gb/s, 1 W
Bluetooth	Bluetooth (Formerly IEEE 802.15.1)	2.4 GHz	WPAN	10–300 m	1 Mb/s–48 Mb/s, 1 w
6LowPAN	IEEE 802.15.4	908.42 MHz/2.4 GHz	WPAN	20–100m	20 Kb/s–250 Kb/s,1 mW
Sigfox	Sigfox	908.42 MHz	LPWAN	<50 km	10–1000 b/s, N/A
LoRaWAN	LoRaWAN	Various	LPWAN	<15 km	0.3–50 Kb/s, N/A
NB-loT	3GPP	180 KHz	LPWAN	<15 km	0–200 Kb/s, N/A
Mobile cellular technology	2G-GSM, GPRS 3G-UMTS, CDMA2000 4G-LTE	865 MHz, 2.4 GHz	GERAN	Entire cellular area	2G: 50–100 kb/s 3G: 200 kb/s 4G: 0.1–1 Gb/s, 1 W
Zigbee	IEEE 802.15.4	2400–2483.5 MHz	Mesh	0–10 m	250 Kbps, 1 mW
NFC	ISO/IEC 13157	13.56 MHz	Point to Point	0.1m	424 Kbps, 1–2 mW

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
