# Peer review of "State-of-the-Art Internet of Things in Protected Agriculture"

_sensors, 2019, doi:10.3390/s19081833_

Reviewer 1 Report

This manuscript has some serious flaws, which prevent it from being published without an extensive revision:

1) Protected agriculture is never defined, nor motivated, nor introduced.

2) A direct consequence of (1) is the lack of motivation and context for a survey on IoT for protected agriculture. Is there a difference from IoT for agriculture and IoT for protected agriculture? If so, which one? And, in which way it influences the motivation and the content of this manuscript?

3) Generally speaking, the IoT-related content is too old. For example, what is the motivation for a survey in wired technologies in 2019? And RFID? And WSN? These are old technologies and there is no need of "yet another" survey about them.

Specific comments:

4) Protected agriculture is not defined in the abstract

5) Line 49 mentions protected agriculture, but as if it were already known and its motivation already provided to the reader. However, this is not the case.

6) IoT for protected agriculture: line 55-57: "With development of Radio Frequency Identification (RFID) technology, agricultural sensor and wireless network, the Internet of Things (IoT) technology has emerged and is gradually being promoted and applied in the protected agriculture field [6].". The problem is that reference [6] is from 2010, i.e., almost 10 years old.

7) Section 2: RFID is not really that important for IoT, even though the term IoT was coined together with this technology. RFID is today more important for other IoT-based applications, such as logistics. Why is it given that prominence in this manuscript?

8) Figure 2: WSNs are in the past and there are lots of papers and survey at least 10 years old in literature. There’s no need to give them there so much attention in a paper about IoT.

9) Section 2: This 3-layered architecture is not really accepted by the community. Please, use well-known references when defining and citing an area so important as IoT. Also, this architecture is too crowded to be useful.

10) Table 1: why a table of hardware boards is important? Where is the motivation?

11) Table 2: why a table with various wired technologies important for IoT? Wired technologies play a significantly smaller role in IoT. Are they more important in protected agriculture? Even so, why a list of old technologies still deserves to be published? In other words, either wired technologies are simply too old to be published or the motivation is weak.

12) Table 3: LPWAN is THE wireless communication technology for IoT. Why is it mentioned only en passant in the middle of other less important technologies?

13) Section 3.3: a section about information processing for IoT should focus in IoT Platforms and Big Data Analytics (machine learning, etc.)

Author Response

Dear reviewer,

Thank you for your comments concerning our manuscript entitled “Internet-of-Things for Precision Agriculture” (sensors-467752). Your comments were highly insightful and enabled us to greatly improve the quality of our manuscript. In the following pages are our point-by-point responses to your comments. Revised portion are highlighted with revised format.

This manuscript has some serious flaws, which prevent it from being published without an extensive revision:

1) Protected agriculture is never defined, nor motivated, nor introduced.

-We agreed with the referee, and we have added the definition of protected agriculture and the motivation.

-You can find the relevant content in page 1 of the revised manuscript.

2) A direct consequence of (1) is the lack of motivation and context for a survey on IoT for protected agriculture. Is there a difference from IoT for agriculture and IoT for protected agriculture? If so, which one? And, in which way it influences the motivation and the content of this manuscript?

-We agreed with the referee, and we have added motivation and context for a survey on IoT for protected agriculture. There are difference from IoT for agriculture and IoT for protected agriculture. The protected agriculture is just a part of the whole agriculture. Some of IoT technology can apply in agriculture but can’t in protected agriculture. For example, the RS technology can be used for estimate crop yields in open-field agriculture. However, it can’t play a role in protected agriculture. We just review the IoT applications in protected agriculture and related technology.

-You can find the relevant content in Introduction of the revised manuscript.

3) Generally speaking, the IoT-related content is too old. For example, what is the motivation for a survey in wired technologies in 2019? And? And WSN? These are old technologies and there is no need of "yet another" survey about them.

- We agreed with the referee, and we have deleted the content of wired technologies and RFID.

4) Protected agriculture is not defined in the abstract

-We agreed with the referee, and we have added the definition of protected agriculture

- You can find the relevant content in abstract of the revised manuscript

5) Line 49 mentions protected agriculture, but as if it were already known and its motivation already provided to the reader. However, this is not the case.

-We agreed with the referee, and we have added the definition of protected agriculture

- You can find the relevant content in Introduction of the revised manuscript

6) IoT for protected agriculture: line 55-57: "With development of Radio Frequency Identification (RFID) technology, agricultural sensor and wireless network, the Internet of Things (IoT) technology has emerged and is gradually being promoted and applied in the protected agriculture field [6].". The problem is that reference [6] is from 2010, i.e., almost 10 years old.

-We agreed with the referee, and we have revised the content and quoted the latest literature.

- You can find the relevant content in Introduction of the revised manuscript

7) Section 2: RFID is not really that important for IoT, even though the term IoT was coined together with this technology. RFID is today more important for other IoT-based applications, such as logistics. Why is it given that prominence in this manuscript?

-We agreed with the referee, and we have deleted the content

8) Figure 2: WSNs are in the past and there are lots of papers and survey at least 10 years old in literature. There’s no need to give them there so much attention in a paper about IoT.

-We agreed with the referee, and we have revised the content and quoted the latest literature.

9) Section 2: This 3-layered architecture is not really accepted by the community. Please, use well-known references when defining and citing an area so important as IoT. Also, this architecture is too crowded to be useful.

- We agreed with the referee, and we have revised the content.

- You can find the relevant content in Page 7 -8 of the revised manuscript

10) Table 1: why a table of hardware boards is important? Where is the motivation?

-We agreed with the referee, and we have deleted the content

11) Table 2: why a table with various wired technologies important for IoT? Wired technologies play a significantly smaller role in IoT. Are they more important in protected agriculture? Even so, why a list of old technologies still deserves to be published? In other words, either wired technologies are simply too old to be published or the motivation is weak.

-We agreed with the referee, and we have deleted the content

12) Table 3: LPWAN is THE wireless communication technology for IoT. Why is it mentioned only en passant in the middle of other less important technologies?

-We agreed with the referee, we have added related content of LPWAN.

- You can find the relevant content in Page 13 of the revised manuscript

13) Section 3.3: a section about information processing for IoT should focus in IoT Platforms and Big Data Analytics (machine learning, etc.)

-We agreed with the referee, we have added related content of Big Data and machine learning.

- You can find the relevant content in Page 14-15 of the revised manuscript.

Reviewer 2 Report

This article is a survey paper that gives reviews of agricultural IoT. The key technologies of IoT in protected agriculture are classified into three categories, namely sensor, data transmission, and information processing. Five applications, including greenhouse environment monitoring and control, crop disease and pest management, livestock farming management, aquaculture management, and agri-food supply chain traceability, are introduced. In addition, different challenges are discussed. There are 147 references in this manuscript. Thus, for a survey paper, I think the coverage of discussion in protected agriculture is enough. I still have some comments in the following.

Since this is a survey paper, all the technique development should be discussed. I suggest the authors don’t need to emphasize what countries the work was presented in.

The readers should have related technique background. I suggest the authors give more detailed discussion on technology, instead of showing the technical terms. For example, in line 230, ANN and SVM are introduced, a good and real reference work should be illustrated in details so that readers could know why these techniques are important and helpful in protected agriculture.

The discussion of applications should be reorganized. The discussion of existing work should be introduced according to their properties. For example, Sections 4.1 and 4.2 introduce the work about plants, Sections 4.3 and 4.4 introduce that about animals (livestock and aquaculture), and Section 4.5 introduces that for food and products. So, I suggest the authors to use three sections to discuss the above three categories. Every section is divided into several subsections based on the issues.

The computing architecture design is also an important issue. However, only three technologies of IoT in protected agriculture, including sensor, data transmission, and information processing, are addressed in this manuscript. The related descriptions of computing architecture design are few and integrated into the introductions of the above technologies. I suggest the authors to add a new section to introduce this issue.

Author Response

Dear reviewer,

Thank you for your comments concerning our manuscript entitled “Internet-of-Things for Precision Agriculture” (sensors-467752). Your comments were highly insightful and enabled us to greatly improve the quality of our manuscript. In the following pages are our point-by-point responses to your comments. Revised portion are highlighted with revised format.

This article is a survey paper that gives reviews of agricultural IoT. The key technologies of IoT in protected agriculture are classified into three categories, namely sensor, data transmission, and information processing. Five applications, including greenhouse environment monitoring and control, crop disease and pest management, livestock farming management, aquaculture management, and agri-food supply chain traceability, are introduced. In addition, different challenges are discussed. There are 147 references in this manuscript. Thus, for a survey paper, I think the coverage of discussion in protected agriculture is enough. I still have some comments in the following.

1 Since this is a survey paper, all the technique development should be discussed. I suggest the authors don’t need to emphasize what countries the work was presented in.

-We agreed with the referee, we have revised the content.

2) The readers should have related technique background. I suggest the authors give more detailed discussion on technology, instead of showing the technical terms. For example, in line 230, ANN and SVM are introduced, a good and real reference work should be illustrated in details so that readers could know why these techniques are important and helpful in protected agriculture.

-We agreed with the referee, we have revised the content.

3) The discussion of applications should be reorganized. The discussion of existing work should be introduced according to their properties. For example, Sections 4.1 and 4.2 introduce the work about plants, Sections 4.3 and 4.4 introduce that about animals (livestock and aquaculture), and Section 4.5 introduces that for food and products. So, I suggest the authors to use three sections to discuss the above three categories. Every section is divided into several subsections based on the issues.

-We agreed with the referee, we have revised the content.

-You can find the relevant content in Page 16-19 of the revised manuscript.

4) The computing architecture design is also an important issue. However, only three technologies of IoT in protected agriculture, including sensor, data transmission, and information processing, are addressed in this manuscript. The related descriptions of computing architecture design are few and integrated into the introductions of the above technologies. I suggest the authors to add a new section to introduce this issue.

-We agreed with the referee, we have added the content about computing architecture.

-You can find the relevant content in Page 16-19 of the revised manuscript.

Reviewer 3 Report

Ensuring food safety is one of the main task posed to the modern world. It is not possible to give up technical progress in striving to provide the public with enough food. one of the tools to improve the efficiency of modern agriculture is to use the opportunities offered by the Internet.

The authors of the article “State of the art Review for Internet of Thing in Protected Agriculture” presented in a synthetic and comprehensive way the possibilities of using the Internet of Thing in protected agriculture. One of the strongest aspects of this study is that the authors also presented the limitations and challenges of the technological solutions described in article.

After reading the article, I formulated the following substantive and editorial comments:

1)   Substantive comments:

Introduction, 38-39 lines: In my opinion the statement "get rid of dependence on natural environment" is too categorical. It seems that the attempt to reduce the impact of the natural environment on agricultural production is closer to reality.

Introduction, 75-82 lines: I suggest to devote more attention to defining the purpose of your research and to the definition of verified research hypodezes. In the case of the subject taken by Authors, it seems that the indication of recipients and the manner of using the presented knowledge would also be justified here.

Introduction, 52 line: The reasonable use of IoT in China has a great importance not only for the community of that country, but for the whole world. Due to the area occupied by China, the solutions implemented in the agriculture of this country have an impact on global climate change. This influence is incomparably greater than the
Netherlands or Japan.

Introduction: I know that in the one of the last chapter Authors put a great attention on weekneses and chalange of IoT technology. But for me it will would be good if in the introduction such issues related to the use of modern technologies as: costs of their implementation, their impact on the structure of agricultural production, ethical problems, and of course not only positive environmental problems, safe treacebility would be considered.

Other challenges, 566-586 lines: it seems that the authors could point out at this point such economic consequences of introducing new technologies as concentration of agricultural production, concentration of capital, change in labor demand, the need to retrain some agricultural workers, etc.

2)   Editorial comments:

inconsistency in the use of capital letters in nomenclature: Figure 1, Figure 3,

unnecessarily used capital letter: line 60, 487, 536 for example

unnecessarily used a small letter: line 117, 120 for example

no description of the source of information: Figure 1, Figure 2, Table 1 ……

no spaces: lone 455, 457, 458, 474 for example,

inconsistent typeface: lines 461-464,

non-uniform format of bibliographical descriptions in references: line 663,692, 679, 727, 786 for example.

I hope that my remarks will be helpful.

Author Response

Dear reviewer

Thank you for your comments concerning our manuscript entitled “Internet-of-Things for Precision Agriculture” (sensors-467752). Your comments were highly insightful and enabled us to greatly improve the quality of our manuscript. In the following pages are our point-by-point responses to your comments. Revised portion are highlighted with revised format.

Ensuring food safety is one of the main tasks posed to the modern world. It is not possible to give up technical progress in striving to provide the public with enough food. One of the tools to improve the efficiency of modern agriculture is to use the opportunities offered by the Internet.

The authors of the article “State of the art Review for Internet of Thing in Protected Agriculture” presented in a synthetic and comprehensive way the possibilities of using the Internet of Thing in protected agriculture. One of the strongest aspects of this study is that the authors also presented the limitations and challenges of the technological solutions described in article.

After reading the article, I formulated the following substantive and editorial comments:

1) Substantive comments:

Introduction, 38-39 lines: In my opinion the statement "get rid of dependence on natural environment" is too categorical. It seems that the attempt to reduce the impact of the natural environment on agricultural production is closer to reality.

-We agreed with the referee, we have revised the content

Introduction, 75-82 lines: I suggest to devote more attention to defining the purpose of your research and to the definition of verified research hypodezes. In the case of the subject taken by Authors, it seems that the indication of recipients and the manner of using the presented knowledge would also be justified here.

-We agreed with the referee, we have revised the content

-You can find the relevant content in Page 1 of the revised manuscript.

Introduction, 52 line: The reasonable use of IoT in China has a great importance not only for the community of that country, but for the whole world. Due to the area occupied by China, the solutions implemented in the agriculture of this country have an impact on global climate change. This influence is incomparably greater than theNetherlands or Japan.

-We agreed with the referee.

Introduction: I know that in the one of the last chapter Authors put a great attention on weekneses and challange of IoT technology. But for me it will would be good if in the introduction such issues related to the use of modern technologies as: costs of their implementation, their impact on the structure of agricultural production, ethical problems, and of course not only positive environmental problems, safe treacebility would be considered.

Other challenges, 566-586 lines: it seems that the authors could point out at this point such economic consequences of introducing new technologies as concentration of agricultural production, concentration of capital, change in labor demand, the need to retrain some agricultural workers, etc.

-We agreed with the referee, we have revised the content

-You can find the relevant content in Page 28-27 of the revised manuscript.

2)   Editorial comments:

inconsistency in the use of capital letters in nomenclature: Figure 1, Figure 3,

-We agreed with the referee, we have revised the content

unnecessarily used capital letter: line 60, 487, 536 for example

-We agreed with the referee, we have revised the content

unnecessarily used a small letter: line 117, 120 for example

-We agreed with the referee, we have revised the content

no description of the source of information: Figure 1, Figure 2, Table 1 …

-We agreed with the referee, we have revised the content.

no spaces: lone 455, 457, 458, 474 for example,

-We agreed with the referee, we have revised the content.

inconsistent typeface: lines 461-464,

-We agreed with the referee, we have revised the content.

non-uniform format of bibliographical descriptions in references: line 663,692, 679, 727, 786 for example.

I hope that my remarks will be helpful.

-We agreed with the referee, we have revised the content.

Reviewer 4 Report

The topic of the manuscript is very promising and interesting and in principle, it is within the specific scope of the Journal. However, the manuscript is not sufficiently organised as a review article.

The review article should be focused better on the object of the review.

The Abstract is not adequately organized: it should describe better the context and intention of the review, provide a general picture of the methodological approach, describe the main outcomes, and conclusions.

Keywords should are poor and insufficient for indexing and searching. You cannot use sentences as keywords.

Introduction is not much clear and it does not describe clearly the subject reviewed. The introduction should be organized better, to provide information about the context, to indicate the motivation for the review, to define the research question and to explain the text structure. Moreover, in the Introduction the protected agriculture should be defined.

Some information on Materials and Methods should be provided (data sources, search terms, search strategy, selection criteria, etc.) to enable motivated researchers to repeat the review.

The main part of the review article requires a coherent structure to develop the section structure.

Conclusions should answer the research question set in the introduction and the implications of the findings. In addition, conclusions should provide interpretations by Authors and identify the unsolved questions, if any. Consequently, the future perspectives of research should follow the conclusions and to be connected particularly to the unsolved questions.

Author Response

Dear reviewer

Thank you for your comments concerning our manuscript entitled “Internet-of-Things for Precision Agriculture” (sensors-467752). Your comments were highly insightful and enabled us to greatly improve the quality of our manuscript. In the following pages are our point-by-point responses to your the comments. Revised portion are highlighted with revised format.

The topic of the manuscript is very promising and interesting and in principle, it is within the specific scope of the Journal. However, the manuscript is not sufficiently organised as a review article.

The review article should be focused better on the object of the review.

The Abstract is not adequately organized: it should describe better the context and intention of the review, provide a general picture of the methodological approach, describe the main outcomes, and conclusions.

-We agreed with the referee, we have revised the context and intention of the review.

-You can find the relevant content in the Abstract of the revised manuscript.

Keywords should are poor and insufficient for indexing and searching. You cannot use sentences as keywords.

-We agreed with the referee, we have revised the content of the review.

-You can find the relevant content in the Keywords of the revised manuscript.

Introduction is not much clear and it does not describe clearly the subject reviewed. The introduction should be organized better, to provide information about the context, to indicate the motivation for the review, to define the research question and to explain the text structure. Moreover, in the Introduction the protected agriculture should be defined.

-We agreed with the referee, we have revised the content of the review.

-You can find the relevant content in the Introduction of the revised manuscript.

Some information on Materials and Methods should be provided (data sources, search terms, search strategy, selection criteria, etc.) to enable motivated researchers to repeat the review.

-We agreed with the referee, we have revised the content of the review.

-You can find the relevant content in the Introduction of the revised manuscript.

The main part of the review article requires a coherent structure to develop the section structure.

-We agreed with the referee, we have revised the content of the review.

Conclusions should answer the research question set in the introduction and the implications of the findings. In addition, conclusions should provide interpretations by Authors and identify the unsolved questions, if any. Consequently, the future perspectives of research should follow the conclusions and to be connected particularly to the unsolved questions.

-We agreed with the referee, we have revised the content of the review.

-You can find the relevant content in the Conclusions of the revised manuscript.

Round  2

Reviewer 1 Report

Authors answered my key concerns in this revised version.

However, now that this manuscript looks like a survey, I have some more concerns:

1) IoT and WSN are in some places treated as they were synonyms, but they are not. Many researchers do this in order to upgrade their research in WSN and pass it as IoT. However, this is not scientifically correct, since IoT does not need a mesh-style WSN with power-based routing, where one node forwards packets of other nodes. As a matter as fact, with the rise of LPWAN, a clumsy mesh-based WSN is not needed anymore.

2) This manuscript should focus more on LPWAN in a general way.

3) English must be improved, as there are still some typos. Also, authors use the article "the" to refer to "the IoT", which is not usual. "the Internet of Things" is ok, but not "the IoT". IoT used without article.

Author Response

Dear reviewer,

Thank you for your comments concerning our manuscript entitled “Internet-of-Things for Precision Agriculture” (sensors-467752). Your comments were highly insightful and enabled us to greatly improve the quality of our manuscript. In the following pages are our point-by-point responses to your comments. Revised portion are highlighted with revised format.

Authors answered my key concerns in this revised version.

However, now that this manuscript looks like a survey, I have some more concerns:

1) IoT and WSN are in some places treated as they were synonyms, but they are not. Many researchers do this in order to upgrade their research in WSN and pass it as IoT. However, this is not scientifically correct, since IoT does not need a mesh-style WSN with power-based routing, where one node forwards packets of other nodes. As a matter as fact, with the rise of LPWAN, a clumsy mesh-based WSN is not needed anymore.

-We agreed with the referee, we have revised the content.

-You can find the relevant content in Page 11 of the revised manuscript.

2) This manuscript should focus more on LPWAN in a general way.

-We agreed with the referee, we have revised the content.

-You can find the relevant content in Page 8, 9, 11, 15, 17 of the revised manuscript.

3) English must be improved, as there are still some typos. Also, authors use the article "the" to refer to "the IoT", which is not usual. "the Internet of Things" is ok, but not "the IoT". IoT used without article.

-We agreed with the referee, we have revised the content.

-You can find the relevant content in the revised manuscript.

Reviewer 2 Report

Most of my comments have been addressed well. I have one more comment. Since Figure 1 was acquired from the work [16], so that its original work should be cited, similar to Figure 5. The authors should check all figures in this manuscript carefully.

Author Response

Dear reviewer,

Thank you for your comments concerning our manuscript entitled “Internet-of-Things for Precision Agriculture” (sensors-467752). Your comments were highly insightful and enabled us to greatly improve the quality of our manuscript. In the following pages are our point-by-point responses to your comments. Revised portion are highlighted with revised format.

Most of my comments have been addressed well. I have one more comment. Since Figure 1 was acquired from the work [16], so that its original work should be cited, similar to Figure 5. The authors should check all figures in this manuscript carefully.

-We agreed with the referee, we have revised the content.

-You can find the relevant content in Page 3-5 of the revised manuscript.